# Controlling Woody Weed Chinese Elm (*Celtis sinensis* Pers.) with Stem-Implanted Herbicide Capsules

**DOI:** 10.3390/plants11030444

**Published:** 2022-02-06

**Authors:** Ciara Jade O’Brien, Vincent Mellor, Victor Joseph Galea

**Affiliations:** School of Agriculture and Food Sciences, Gatton Campus, The University of Queensland, Gatton, QLD 4343, Australia; ciara.obrien@uqconnect.edu.au (C.J.O.); vam103@maths.uq.edu.au (V.M.)

**Keywords:** Chinese elm, woody weed, weed management, chemical control, stem implantation

## Abstract

Chinese elm [*Celtis sinensis* Pers.] is an emerging environmental weed naturalised throughout the coastal and riparian (creek-banks, river margins, and streams) regions of eastern Australia. Throughout this introduced range, its management is limited to the application of synthetic herbicides and mechanical clearing operations (terrain and soil type permitting). The current mechanisms of chemical control (basal bark spraying, stem-injection, and cut-stump applications) often result in collateral damage to non-target native species (such as *Eucalyptus* spp. and *Casuarina cunninghamiana* Miq.) through herbicidal drift, runoff or leaching into adjacent habitats. This has raised concerns regarding the suitability of synthetic herbicides in ecologically sensitive (e.g., riparian zones, rainforest margins, and woodlands) or low-value habitats, thereby promoting significant developments in the fields of integrated weed management. This study investigated the effectiveness of a novel stem-implantation system for controlling woody weed species in the context of a conserved habitat. A replicated trial (n = 315) was established among a naturally occurring population of *C. sinensis*. This trial involved the mapping, measurement, and treatment of this invasive species with five encapsulated synthetic herbicides, as well as an untreated control and benchmark treatment (diesel + Access^TM^). A significant effect (*p* < 0.05) on plant vigour and functional canopy was discerned for each assessment period following trial establishment. The highest incidence of mortality was observed among the individuals treated with glyphosate (245 mg/capsule), aminopyralid and metsulfuron-methyl (58.1 and 37.5 mg/capsule) and picloram (10 mg/capsule), achieving a similar response to the basal bark application of diesel and Access^TM^ (240 g/L triclopyr, 120 g/L picloram, and 389 g/L liquid hydrocarbon). This was also evidenced by a rapid reduction in functional canopy (i.e., no or little living leaf tissue) from three weeks after treatment. Unlike their industry counterparts, these encapsulated herbicides are immediately sealed into the vascular system of the target species by a plug. This significantly minimises the possibility of environmental or operator exposure to synthetic compounds by providing a targeted, readily calibrated herbicide application.

## 1. Introduction

Chinese elm (*Celtis sinensis* Pers.) is a deciduous or semi-deciduous tree native to the slopes of eastern Asia, most notably China, Korea, Taiwan and Japan [1,2]. However, this species has spread from its endemic habitat to the coastal and sub-coastal regions of Australia, New Zealand and South Africa through its deliberate introduction as an ornamental plant [2]. In Australia, its naturalisation throughout the riparian zones (creek-banks, river margins, and streams) of south-eastern Queensland (Brisbane, Nambour, Toowoomba, Dalby) and north-eastern New South Wales (Lismore, Kyogle, Tweed Harbour, Coffs Harbour) has caused the displacement of existing native vegetation, thereby threatening the biodiversity, resilience, and integrity of natural ecosystems [1,2,3,4]. This adversely affects populations of resident fauna (e.g., Koala *Phascolarctos cinereus*, Common Brushtail Possum *Trichosurus vulpecula*, Greater Glider *Petauroides volans*, Rufous Rat-Kangaroo *Aepyprymnus rufescens*, and Black Wallaby *Wallabia bicolor*) by altering habitat conditions or resource availability (foliage, seeds, nectar, or sap) [1,4,5,6]. It has also formed dense infestations in disturbed sites such as urbanised bushlands, parklands and roadsides [1,3]. Given this evidence of invasiveness, *C. sinensis* is ranked in the ten highest invasive species in south-east Queensland alongside other notable woody weeds such as Lantana (*Lantana camara* var. *camara*), Camphor Laurel (*Cinnamomum camphora*) and Broad-Leaf Pepper Tree (*Schinus terebinthifolius*) [7].

The management of *C. sinensis* is currently limited to the application of synthetic herbicides and mechanical clearing operations (terrain and soil type permitting). The manual removal of individual plants may be practical for the initial clearing of higher-density (>150 plants/ha) or isolated infestations [4]. This can be achieved through the hand pulling of small seedlings (height < 30 cm), bulldozing or controlled grazing [4,5,8]. However, these manual attempts at control are largely ineffective due to the vigorous resprouting capacity of severed plants [4,5].

Although the herbicides registered for its management are limited, the minor use of agricultural and veterinary (AGVET) chemical products is authorised under a permit (APVMA Permit PER11463) for the control of environmental weeds in non-agricultural areas [5,9]. In particular, the cut-stump or basal application of synthetic auxin chemicals (i.e., fluroxypyr, triclopyr or picloram) is recommended with compliance to label directions and permit conditions [5,8,9]. The latter is performed at the base of the target species (plants with <20 cm basal diameter) with a mixture of oil-soluble herbicide and diesel distillate to assist penetration through the bark [9,10]. This has been proven effective for the management of scattered, lower-density infestations of parkinsonia (*Parkinsonia aculeata* L.) [11], mimosa (*Mimosa pigra* L.) [12], mesquite (*Prosopis* L. species) [13], bellyache bush (*Jatropha gossypiifolia* L.) [14], calotrope (*Calotropis procera*) [15], yellow oleander (*Cascabela thevetia* (L.) Lippold) [9] and white weeping broom (*Ratama raetam* (Forssk.) Webb) [16] in the Australian landscape. For larger woody weeds, the cut-stump method involves the painting or spraying of herbicide to the exposed surface of a felled stump [10]. (Whilst their efficacy is undisputed, there are concerns regarding the suitability of these application methods in ecologically sensitive (e.g., riparian zones, rainforest margins, and woodlands) or low-value habitats [10,17].) The imprecise or excessive application of herbicides may result in collateral damage to non-target native species (such as *Eucalyptus* spp. and *Casuarina cunninghamiana* Miq.) through herbicidal drift, runoff or leaching into adjacent habitats [10,17]. The movement of herbicide between the anastomosed roots of neighbouring plants has also been documented [10,18] with conventional stem-injection methods such as ‘frill and fill’ [19] or ‘drill and fill’ [9]. This greater appreciation for environmental stewardship has promoted significant developments in the field of woody weed management by reducing dosage or improving application methods [9,10].

This study investigated the effectiveness of BioHerbicides Australia’s (www.bioherbicides.com.au) proprietary stem-implantation system (InJecta 800^®^) and Di-Bak^®^ range of synthetic herbicides for controlling *C. sinensis* in the context of a conserved habitat. This lightweight, handheld device was initially developed for the encapsulated delivery of three endophytic fungal species (*Lasiodiplodia pseudotheobromae*, *Macrophomina phaseolina*, *Neoscytalidium novaehollandiae*) for the management of parkinsonia (*P. aculeata*) on Australian rangelands [20,21]. This novel technology has since been expanded for the application for other endophytic organisms, as well as synthetic compounds (herbicides, fungicides, and insecticides) available in dry formulations [10,20,21]. More recently, the synthetic herbicide formulations have been trialed against a range of woody weed species such as parkinsonia (*P. aculeata*), prickly acacia (*Vachellia nilotica*), leucaena (*Leucaena leucocephala*), camphor laurel (*Cinnamomum camphora*) and privet (*Ligustrum lucidum*) [10,21,22]. These studies demonstrated that encapsulated glyphosate (~350 mg per capsule) was highly efficacious against all species except for parkinsonia (*P. aculeata*) [10]. Other formulations under evaluation include metsulfuron-methyl, picloram and imazapyr [10]. Unlike its industry counterparts, these encapsulated synthetic herbicides are immediately sealed into the target species, thereby minimising the possibility of unintentional chemical exposure to the neighbouring native vegetation or human operator [10].

## 2. Results

### 2.1. Weather Data

A record of monthly rainfall (mm) from January 2018 to March 2020 was retrieved from Old Hidden Vale Station, Grandchester [23] (Figure 1). A significant rainfall event (180 mm) was recorded during the establishment of the trial in mid-March 2019. Although this corresponds with the wet season (November to April) in Queensland, the intensity of rainfall was greater than the previous (2018) and subsequent year (2020). The following nine months (April 2019 to December 2019) were unusually dry with a total rainfall of 80.5 mm relative to the 312 mm of rainfall recorded in the previous year (2018). However, the rainfall returned to expected levels in the latter months of the wet season (January, February, and March of 2020).

### 2.2. Encapsulated Synthetic Herbicide Trial

A significant effect (*p* < 0.05) on plant vigour was discerned for each assessment period following trial establishment (week 3, 8, 15, 20, 25, 35, and 52) (Table 1). The benchmark treatment (diesel + Access^TM^) and glyphosate had the most immediate effect on plant vigour, whereby all treated individuals were deemed ‘dead’ (stress score of three) or ‘dying’ within fifteen weeks (Table 1, Table 2). A similar trend in plant mortality was observed with aminopyralid and metsulfuron-methyl and picloram, as evidenced by their steadily increasing stress scores (Table 1). However, there was no significant difference (*p* > 0.05) between these four treatments at the conclusion of the trial (i.e., week 52) (Table 1, Table 2). There was also a high incidence of mortality (82.22%) among the individuals treated with metsulfuron-methyl (Table 1). However, the transition from being ‘distressed’ (stress score of two) to ‘dead’ (stress score of three) was slower relative to the preeminent treatments (Table 1). Whilst the effect of imazapyr plateaued from week eight to week twenty-five, achieving the lowest degree of mortality (46.67% at week fifty-two) among the encapsulated treatments (Table 1, Table 2). The health of the untreated plants (i.e., control treatment) was unaffected throughout the trial period (Table 2). Hence, a change in the condition of the target species was attributed to the explanatory variable (i.e., synthetic herbicide) rather than an unaccounted-for factor, such as drought stress, nutrient deficiency or plant disease.

Similarly, a significant effect *p* < 0.05 was discerned for functional canopy at each assessment period weeks 3, 8, 15, 20, 25, 35, and 52 Table 3. This value is referring to the aboveground portion of the plant with photosynthetic capacity i.e., healthy, living foliage. The benchmark treatment diesel + Access^TM^ and glyphosate caused a rapid reduction in functional canopy, whereby no living tissue 0% was remaining at fifteen weeks after treatment Table 3, Figure 2. A similar downward trend in functional canopy was also observed with aminopyralid and metsulfuron-methyl and picloram, as shown in Table 3. However, there was no significant difference *p* > 0.05 between these four treatments from week twenty onwards in terms of foliage loss or functional canopy Table 3, Figure 2. The individuals treated with metsulfuron-methyl and imazapyr experienced a more progressive, steady reduction in functional canopy Table 3. Despite there being no living tissue 0% remaining at week twenty, their canopies recovered slightly at the conclusion of the trial Table 3, Figure 2. This may be epicormic growth in response to herbicidal injury or distress rather than a flashback attempt. The untreated plants i.e., control were also affected between week twenty and week thirty-five Table 3, Figure 2. This is characteristic behaviour in the autumn March, April, and May and winter June, July, and August months i.e., prolonged dry conditions given the deciduous and semi-deciduous nature of this tree species [24]. The condition of the untreated plants was restored following consistent rainfall in the summer months January, February, and March Table 3.

## 3. Discussion

The result of this study suggests that the successful management of *C. sinensis* in conserved habitats (e.g., riparian zones, woodlands, and rainforest margins) is possible through the implantation of encapsulated synthetic herbicides. The highest incidence of mortality was observed among the individuals treated with glyphosate (245 mg/capsule), aminopyralid plus metsulfuron-methyl (58.1 and 37.5 mg/capsule) and picloram (10 mg/capsule), achieving a similar response to the industry accepted standard (i.e., basal bark application of diesel + Access^TM^). This is evidenced by their rapidly increasing stress scores that translated to entirely (100%) ‘dead’ *C. sinensis* plants by the conclusion of the trial. Other symptoms of herbicidal injury were also apparent such as the puckering, longitudinal cracking and bleaching of the outer bark tissue. Despite causing considerable distress, the least effective synthetic treatments were metsulfuron-methyl (198 mg/capsule) and imazapyr (262.5 mg/capsule) on a comparative basis. The health of the untreated plants was unaffected (0% mortality) throughout the trial period. However, a slight reduction in functional canopy was recorded from week twenty to week thirty-five. This is characteristic behaviour in the autumn and winter months (i.e., prolonged dry conditions) given the deciduous and semi-deciduous nature of this species [24]. As expected, the condition of the untreated plants was restored following consistent rainfall (247.5 mm cumulative) in the summer months (January, February, and March).

There are a limited number of herbicides registered by the Australian Pesticides and Veterinary Medicines Authority (APVMA) for the management of *C. sinensis*. However, the use of agricultural and veterinary (AGVET) chemical products is authorised in non-agricultural areas under an off-label use permit (APVMA Permit PER11463) [5]. In particular, the basal bark application of fluroxypyr (200 g/L, 333 g/L or 400 g/L) or triclopyr (240 g/L) plus picloram (120 g/L) with diesel distillate is recommended for the treatment of saplings or regrowth with a basal diameter of ≤20 cm [3,5,25]. The latter (triclopyr + picloram) is also registered for the management of other woody weed species such as parkinsonia (*Parkinsonia aculeata* L.), prickly acacia (*Vachellia nilotica*), lantana (*Lantana camara*), leucaena (*Leucaena leucocephala*), mimosa bush (*Mimosa pigra* L.), camphor laurel (*Cinnamomum camphora*) and brigalow (*Acacia harpophylla*) [22,25]. This supported its selection as the benchmark treatment for this study. Unsurprisingly, the benchmark treatment caused a rapid deterioration in plant health (100% mortality at fifty-two weeks). However, there was no significant difference (*p* > 0.05) between the most efficacious encapsulated synthetic herbicides (glyphosate, aminopyralid + metsulfuron-methyl, picloram) and the benchmark treatment at the conclusion of the trial. This infers that the stem-implantation technology meets the current industry standard.

The delivery of encapsulated synthetic herbicides has been proven successful for the management of other woody weed species such as prickly acacia (*V. nilotica*), leucaena (*L. leucocephala*), mimosa bush (*M. pigra*), camphor laurel (*C. camphora*) and privet (*Ligustrum lucidum*) [10,21,22]. These trials followed the guidelines of the Australian Pesticides and Veterinary Medicines Authority (APVMA) for research on pesticide efficacy [21]. Typically, this efficacy criterion requires (≥) fifteen plants per treatment group with a minimum of three replications in a randomized complete block design (RCBD) [21]. These studies demonstrated that encapsulated glyphosate (~350 mg/capsule) was highly efficacious against all species except for parkinsonia (*P. aculeata*) [10,21]. The effectiveness of stem-injected (i.e., ‘drill and fill’ and ‘frill and fill’) glyphosate has also been documented in mimosa bush (*M. pigra*), yellow oleander (*Cascabela thevetia*) and tree of heaven (*Ailanthus altissima*) [9,22,26]. The results of this study support these previous findings that encapsulated glyphosate (~245 mg/capsule) is a promising candidate for woody weed management.

The stem-implantation technology (InJecta^®^) has many benefits by providing a targeted, readily calibrated herbicide application. This methodology delivers a minimum recommended lethal dose of chemical directly into the vascular system of the target species, thereby fully capturing (100%) the active agent internally [10,21]. The possibility of environmental or operator exposure (through dermal absorption or respiratory inhalation) to synthetic compounds is greatly reduced compared to its industry counterparts, such as canopy application or stem spraying [21]. In our study, there was no visual indication of herbicidal injury or distress (e.g., distorted growth, foliage loss, interveinal chlorosis, and necrosis) [27] among the untreated *C. sinensis* plants or adjacent non-target vegetation. This indicates that little or no translocation occurred between treated and untreated plants. Hence, this method is deemed broadly appropriate for the management of woody weed species in ecologically sensitive habitats (e.g., riparian zones, rainforest margins, national parks, woodlands, and wetlands).

Despite being highly efficacious, these encapsulated chemical formulations also contain less (20% to 30%) herbicidal active ingredients relative to other control options (e.g., basal bark spraying, cut stump, and ‘drill and fill’) [10]. For example, a single dose (1 mL) of undiluted RoundUp^®^ has 360 mg of active ingredient (glyphosate) for ‘axe-cut’ applications. Whilst a single glyphosate capsule (size 0 hypromellose capsule) contains 245 mg of active ingredient. The occurrence of ‘flashback’ will be reduced under these lowered dosages [10], as well as the development of herbicide resistance or tolerance in targeted species [28].

The primary intent of this study was to investigate the effectiveness of BioHerbicides Australia’s (BHA Pty Ltd., Brisbane, QLD, Australia) proprietary stem-implantation system and Di-Bak^®^ range of synthetic herbicides for controlling *C. sinensis* in the context of a conserved habitat. It was found that the most effective encapsulated herbicides were glyphosate (245 mg/capsule), aminopyralid and metsulfuron-methyl (58.1 and 37.5 mg/capsules) and picloram (10 mg/capsule), achieving similar degree of plant mortality relative to the benchmark treatment (i.e., basal bark application of diesel + Access^TM^). Unlike its industry counterparts, this novel technology (InJecta^®^) delivers concentrated dry formulations directly into the vascular system of the target species where all (100%) of the active ingredient is captured internally [10,21]. This has the potential for (i) reducing the amount of active agent required, (ii) preventing environmental exposure to plant protection chemicals and (iii) improving operator safety [10,21]. Hence, this methodology could be a replacement for stem-injection or cut-stump applications in ecologically sensitive habitats (riparian zones, rainforest margins, national parks, woodlands, wetlands) [10], as well as for the management of root and stem disorders in plantation crops (e.g., rubber *Hevea brasiliensis* and oil palm *Elaeis guineensis*) [21]. Further research to optimise the dosage level and placement of the most effective treatments (glyphosate, aminopyralid and metsulfuron-methyl) is currently underway.

## 4. Materials and Methods

### 4.1. Experimental Site, Design and Treatments

A replicated trial (n = 315) was established among a naturally occurring population of *C. sinensis* located on the banks of Franklin Vale Creek (near Grandchester, southeast Queensland: 27°44′46″ S, 152°27′17″ E). This trial involved the mapping, measurement and treatment of individual plants with five encapsulated synthetic herbicides sourced from BioHerbicides Australia’s (Bioherbicides Australia Pty Ltd., Brisbane, QLD, Australia) Di-Bak^®^ range of registered and developmental products (Table 4). A control (untreated plants) and benchmark treatment were also included for performance comparison, this being the basal bark application of diesel + Access^TM^ herbicide (240 g/L triclopyr, 120 g/L picloram, and 389 g/L liquid hydrocarbon). 

The trial was established in mid-March 2019 using a randomised complete block design (RCBD) with three blocks. Within each block, the seven treatments were randomly assigned to a total of fifteen plants (of similar age) complying with the recommendations of the Australian Pesticides and Veterinary Medicines Authority (APVMA) for efficacy evaluation on woody weed species. The plants (stem circumference range of ≥15 cm to 90 cm) within each treatment plot were clustered into small groupings or rows along the creekbank. Treatment plots were differentiated from one another by coloured flagging tape, clearly labelled and their respective GPS waypoints determined using a handheld Garmin^®^ 62s GPS device (Garmin Australasia Pty Ltd., Eastern Creek, NSW, Australia). 

### 4.2. Treatment Application

The encapsulated synthetic herbicides were administered via the InJecta^®^ handheld device (Bioherbicides Australia Pty Ltd., Brisbane, QLD, Australia) Figure 3a]. This applicator is powered by a cordless drill using an 8 mm drill bit creating a hole (25 mm depth) into the plant stem at an approximate height of one metre (Figure 3b) [29]. The withdrawal of the drill backwards is followed by the rotation of the magazine, thereby priming a single capsule (21.6 mm × 7.6 mm) containing the dry herbicide formulation and plug for delivery (which are in-tandem within each of the thirty chambers of the magazine) (Figure 3c) [22,29]. The capsule and plug are then simultaneously inserted into the drilled hole through the forward movement of the non-rotating drill [29]. The synthetic herbicide is immediately sealed into the target species by a polypropylene plastic plug (Figure 3d) [22,29]. This exclusion of an oxidizing atmosphere to the wound tissues facilitates the absorption of xylem and phloem fluids by the capsule (i.e., dissolving the herbicide) [29]. 

The applied dosage was determined by the stem circumference of the plant or each branch (multiple-stemmed plant) at chest height. A single capsule was administered for every 15 cm incremental increase in stem circumference. In the case of multiple doses, the capsules were spaced evenly around the plant stem.

The basal application of Access^TM^ (Corteva Agriscience Pty Ltd., Sydney, NSW, Australia) herbicide (240 g/L triclopyr, 120 g/L picloram, and 389 g/L liquid hydrocarbon) with diesel (dilution rate of 1 L/60 L) was achieved with a manual pressure sprayer (Nylex 8 L Heavy Duty Shoulder Sprayer; Ames Australia, Doncaster, Victoria, Australia). The entirety of the stem and root collar area was treated liberally from ground level to an approximate height of 60 cm (as per manufacturer’s instructions) for sufficient penetration through the bark. The appropriate safety equipment (Bossweld elbow-length gloves (Dynaweld, Prestons, NSW, Australia), valved activated carbon respirator (3M Australia Pty Ltd., North Rhyde, NSW, Australia, and covered clothing) was worn during the preparation and application of the solution.

### 4.3. Trial Assessment

The trial was rated at approximate monthly intervals by recording the percentage of foliage loss, the colour composition (percentage green, yellow and brown) of the remaining canopy and the overall vigour of each individual plant. Based upon visual observation, a rating of 0 to 10 (0 = 0%, 1 = 1–10%, 2 = 11–20%, 3 = 21–30%, 4 = 31–40%, 5 = 41–50%, 6 = 51–60%, 7 = 61–70%, 8 = 71–80%, 9 = 81–90%, 10 = 91–99%, 11 = 100%) was given for each plant indicating the percentage (%) of total foliage loss since the establishment of the trial. The overall vigour of each plant was also recorded and expressed as a stress score (1 = healthy, 2 = distressed, 3 = dead). This was discerned by removing the outermost layer of the bark with a rasp to reveal the colour of the tissue beneath. Additionally, an auditory assessment of the degree of hydration was conducted by tapping the stem with a hammer. Other observable symptoms of stress were recorded such as the splitting or discolouration of the bark, sap seepage from the implantation site and insect damage.

The ’functional canopy’ of each plant was also calculated from the percentage (%) of foliage loss and the colour composition of the remaining canopy.$$Functional\;Canopy=\left(\frac{Percentage\;Existing\;Canopy}{100}\right)\;\times \;\left(\frac{Percentage\;Green\;Canopy}{100}\right)$$
This rating refers to the aboveground portion of the tree canopy that is functional, living tissue. This is expressed on a scale of zero to one, whereby a value of one is indicative of a highly functional or healthy canopy (i.e., full, green canopy).

### 4.4. Data Analysis

The treatment effects on stress score and functional canopy were analysed using RStudio^®^ (RStudio Inc., Boston, MA, USA). Although stress score is an ordinal scale (i.e., quantitative data), a one-way analysis of variance (ANOVA) was performed by taking the mean (µ) value from each replicate. The functional canopy was also analysed using the same approach (i.e., one-way ANOVA). All pairwise comparisons among treatment means (µ) were estimated with the emmeans (estimated marginal means) package. 

## Figures and Tables

**Figure 1 plants-11-00444-f001:**
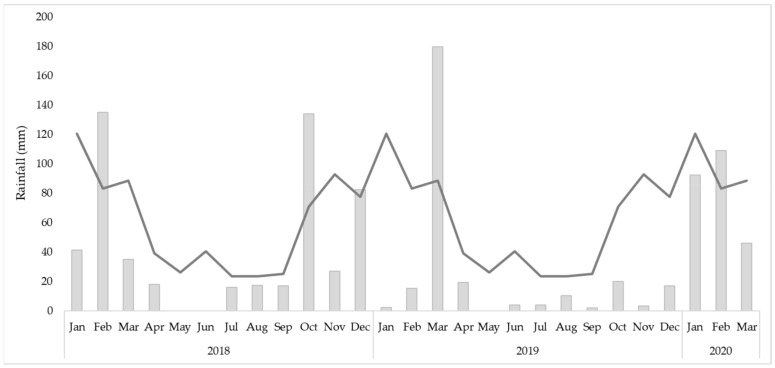
The monthly rainfall (mm) records at Old Hidden Vale Station, Grandchester, Queensland from January 2018 to March 2020. The line is indicative of the long-term (2000–2019) monthly rainfall means (µ).

**Figure 2 plants-11-00444-f002:**
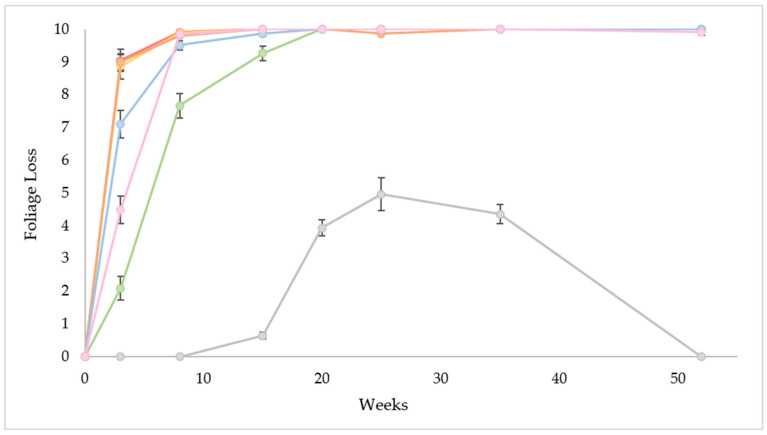
The mean (µ) foliage loss (0 = 0%, 1 = 1–10%, 2 = 11–20%, 3 = 21–30%, 4 = 31–40%, 5 = 41–50%, 6 = 51–60%, 7 = 61–70%, 8 = 71–80%, 9 = 81–90%, 10 = 91–100%) of the six chemical treatments (
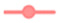
 = diesel + Access^®^; 
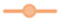
 = glyphosate; 
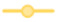
 = picloram; 
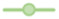
 = imazapyr; 
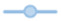
 = aminopyralid + metsulfuron-methyl; 
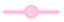
 = metsulfuron-methyl) and the control treatment (
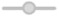
) for each assessment period (week 0, 3, 8, 15, 20, 25, 35 and 52) under field conditions. The error bars represent the standard error (SE).

**Figure 3 plants-11-00444-f003:**
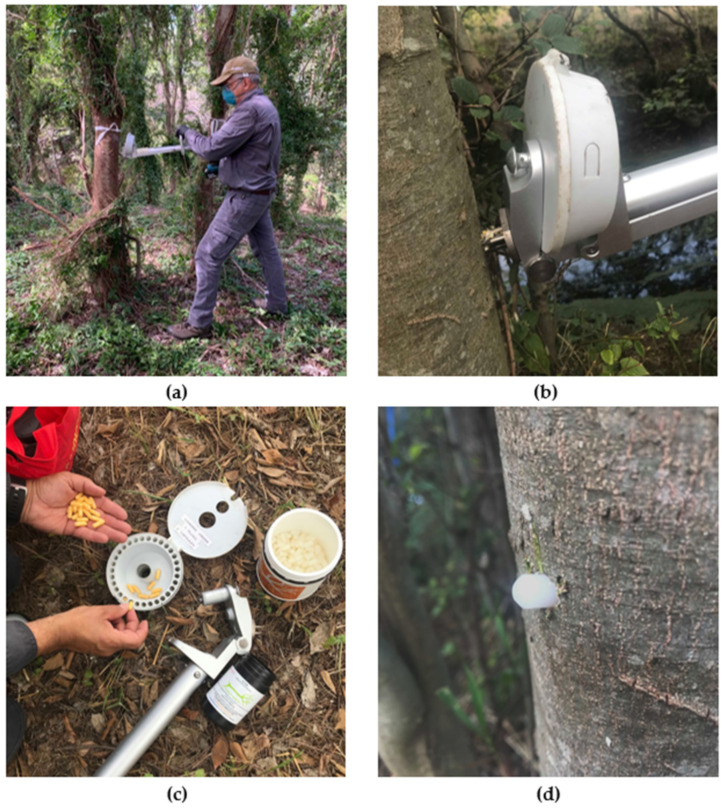
(**a**) Implanting a synthetic herbicide capsule into the stem of a *C. sinensis* plant using the InJecta^®^ handheld device; (**b**) rotating drill bit (8 mm) creating a hole into the plant stem; (**c**) loading the magazine with synthetic herbicide capsules and polypropylene plugs; (**d**) polypropylene plug partially protruding from the implantation site of a treated *C. sinensis* plant.

**Table 1 plants-11-00444-t001:** One-way analysis of variance, estimated marginal means (EMM), and standard error (SE) of stress score for each assessment period (week 0, 3, 8, 15, 20, 25, 35, and 52). The superscript letters (i.e., compact letter displays) denote all pairwise comparisons among treatment means (µ).

	Week															
	0		3		8		15		20		25		35		52	
*p*-Value	0.468		4.70 × 10 ^−5^ ***		1.12 × 10 ^−7^ ***		8.11 × 10 ^−9^ ***		3.02 × 10 ^−9^ ***		2.82 × 10 ^−10^ ***		1.12 × 10 ^−9^ ***		1.35 × 10 ^−12^ ***	
	EMM	SE	EMM	SE	EMM	SE	EMM	SE	EMM	SE	EMM	SE	EMM	SE	EMM	SE
Control	1.00 ^a^	0	1.00 ^b^	0	1.00 ^d^	0	1.02 ^c^	0.022	1.02 ^c^	0.022	1.00 ^c^	0	1.00 ^c^	0	1.00 ^c^	0
Diesel + Access ^TM^	1.00 ^a^	0	1.96 ^a^	0.031	2.87 ^a^	0.051	2.98 ^a^	0.022	2.98 ^a^	0.022	2.96 ^a^	0.031	2.98 ^a^	0.022	3.00 ^a^	0
Glyphosate	1.00 ^a^	0	1.96 ^a^	0.031	2.84 ^a^	0.055	2.87 ^a^	0.051	2.91 ^a^	0.043	2.93 ^a^	0.038	2.98 ^a^	0.022	3.00 ^a^	0
Picloram	1.00 ^a^	0	2.00 ^a^	0	2.56 ^ab^	0.075	2.78 ^a^	0.063	2.80 ^a^	0.060	2.78 ^a^	0.063	2.96 ^a^	0.031	3.00 ^a^	0
Aminopyralid + Metsulfuron-Methyl	1.00 ^a^	0	1.93 ^a^	0.038	2.44 ^abc^	0.075	2.64 ^a^	0.072	2.69 ^a^	0.070	2.71 ^a^	0.068	2.82 ^a^	0.058	2.98 ^a^	0.022
Metsulfuron-Methyl	1.00 ^a^	0	1.98 ^a^	0.022	2.07 ^c^	0.038	2.22 ^b^	0.063	2.20 ^b^	0.060	2.18 ^b^	0.066	2.36 ^b^	0.072	2.82 ^a^	0.058
Imazapyr	1.00 ^a^	0	1.42 ^b^	0.074	2.18 ^bc^	0.058	2.11 ^b^	0.047	2.09 ^b^	0.043	2.13 ^b^	0.051	2.27 ^b^	0.067	2.47 ^b^	0.075

Significance value *** = 0.

**Table 2 plants-11-00444-t002:** Percentage (%) mortality of each encapsulated treatment at the final assessment (week 52).

Treatment	Mortality %
Control	0
Diesel + AccessTM	100
Glyphosate	100
Picloram	100
Aminopyralid + Metsulfuron-Methyl	97.78
Metsulfuron Methyl	82.22
Imazapyr	46.67

**Table 3 plants-11-00444-t003:** One-way analysis of variance, estimated marginal means (EMM), and standard error (SE) of functional canopy for each assessment period week 0, 3, 8, 15, 20, 25, 35, and 52. The superscript letters i.e., compact letter displays denote all pairwise comparisons among treatment means µ. Significance value *** = 0.

	Week															
	0		3		8		15		20		25		35		52	
*p*-Value	-		1.66 × 10 ^−9^ ***		3.58 × 10 ^−13^ ***		1.99 × 10 ^−9^ ***		<2 × 10 ^−16^ ***		7.17 × 10 ^−12^ ***		2.79 × 10 ^−8^ ***		1.42 × 10 ^−15^ ***	
	EMM	SE	EMM	SE	EMM	SE	EMM	SE	EMM	SE	EMM	SE	EMM	SE	EMM	SE
Control	-	0	1.00 ^a^	0	0.99 ^a^	0.003	0.96 ^a^	0.01	0.64 ^a^	0.01	0.53 ^a^	0.05	0.59 ^a^	0.03	1.00 ^a^	0
Diesel + Access ^TM^	-	0	0.06 ^de^	0.029	0.003 ^d^	0.005	0.0002 ^bc^	0	0.0 ^b^	0	0.0 ^b^	0	0.0 ^b^	0	0.0 ^c^	0
Glyphosate	-	0	0.02 ^e^	0.009	0.004 ^cd^	0.004	0.0 ^c^	0	0.0 ^b^	0	0.0 ^b^	0	0.0 ^b^	0.001	0.0 ^c^	0
Picloram	-	0	0.04 ^e^	0.016	0.02 ^cd^	0.005	0.0002 ^bc^	0	0.0 ^b^	0	0.0 ^b^	0.001	0.0 ^b^	0	0.0 ^c^	0
Aminopyralid + Metsulfuron-Methyl	-	0	0.20 ^d^	0.038	0.03 ^c^	0.009	0.005 ^bc^	0.005	0.0 ^b^	0	0.0 ^b^	0	0.003 ^b^	0.002	0.0 ^c^	0.001
Metsulfuron-Methyl	-	0	0.45 ^c^	0.042	0.02 ^cd^	0.004	0.003 ^bc^	0.001	0.0 ^b^	0	0.001 ^b^	0	0.005 ^b^	0.003	0.013 ^b^	0.01
Imazapyr	-	0	0.72 ^b^	0.040	0.22 ^b^	0.034	0.07 ^b^	0.022	0.0 ^b^	0	0.0 ^b^	0	0.006 ^b^	0.01	0.023 ^b^	0.004

**Table 4 plants-11-00444-t004:** Treatment name, active ingredient(s) concentration and dosage (mg/capsule) of the five encapsulated herbicides. All capsules were sourced from BioHerbicides Australia (BHA Pty Ltd.).

Treatment	Active Ingredient(s)	Active Ingredient Concentration (g/kg)	Dose (mg/Capsule)
Di-Bak AM^®^	Aminopyralid + Metsulfuron Methyl	375 & 300	58.1 & 37.5
Di-Bak M^®^	Metsulfuron Methyl	600	198
Di-Bak G^®^	Glyphosate	700	245
Di-Bak I^®^	Imazapyr	750	262.5
Di-Bak P^®^	Picloram	100	10

## Data Availability

Not Applicable.

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
