# Peer review of "Controlling Woody Weed Chinese Elm (*Celtis sinensis* Pers.) with Stem-Implanted Herbicide Capsules"

_plants, 2022, doi:10.3390/plants11030444_

Round 1
Reviewer 1 Report
In this study entitled “Controlling Chinese Elm (Celtis sinensis) with Di-Bak® Stem-2 Implanted Herbicide Capsules” the efficacy of five encapsulated herbicides against the woody weed Celtis sinensis. In this study are presented useful experimental results that deserved to be published after major revision.
Comments
Title: The title of article should be revised as follow: Controlling woody weed Chinese Elm (Celtis sinensis Pers.) with Stem-Implanted Herbicide Capsules
Abstract: The abstract should be revised focused on the results of this study. Moreover, the commercial names of herbicides should be replaced with names of active ingredients and dose in order to be comparable with the results of other studies
Lines 21-22: The text in parenthesis should deleted since is mentioned in Material and Methods.
Introduction: This section needs revision.
Table 1 should be deleted since is published in other study.
Lines 64-74: This paragraph should be revised. The names of commercial products should be deleted. The authors should add more information about the efficacy of herbicides against the weed Chinese elm applied with different methods (foliar spraying, basal bark spraying etc.).
Lines 75-91: The commercial names of herbicides should be deleted.
Lines 83-88: In these lines the authors mentioned that encapsulate glyphosate efficacy had evaluated in other woody weeds. Which the main results from the studies 11, 12, and 13?
Results: The authors presented the same data either on table or figure. The authors should present their data only once and in tables 2 and 3 should present only the results of statistical analysis. Moreover, the results of total foliage loss for each assessment period should be presented in a figure, while the % efficacy of each herbicide at the final assessment should be also provided.
Discussion: This section needs slightly revision. The authors should include results from other studies about the efficacy of evaluated herbicides in other woody weed species.
Lines 186-212: In these lines the authors mentioned that differences between herbicides are due to mode of action. This should supported by results of other studies.
Lines 218-221: The commercial names of herbicides should be deleted.
Material and methods: This is section needs slightly revision.
Line 300: The size of plots should be included in this sub-section.
Line 320: 8mm should be corrected to 8 mm. The same corrections should made in lines 327, 330, 332, and in discussion section.
Lines 331-332: The phrase should be revise included the term basal application.
Lines 344—345: The authors should mention how many plants evaluated in each plot in order to record the stress score of each treatment. The some information should be mention for the other measurements of this study. The term stress score should also mentioned in this part since is reported several times in Results.
Author Response
Please refer to attached cover letter, thankyou for your constructive review of our mansucript.

Reviewer 2 Report
The approach is innovative and the paper very interesting. The introduction is adequate, the results clear and the discussion sound. Two things could be further improved: a) to add some more scientific measurements since their majority is based in visual observation and b) to highlight more some future uses of the method and its novelty. After these, the paper can be accepted for publication
Author Response

(The authors gave the same response as above.)

Reviewer 3 Report
Review of the manuscript plants-1551053
The manuscript is just a simple observation of the reaction of Celtis sinensis to different herbicides enclosed in stem-implanted capsules. It is interesting as an innovative and environment friendly technology however plant reaction is not deeply studied. As the authors admitted (lines 268-269) these results were preliminary. Therefore I suggest to make more research and resubmit the manuscript.
Other concerns are listed below:
- The same results are presented twice (Figure 2 and Table 2 as well as Table 3 and Figure 3). The authors should choose only tables (eventually add SE values) or present only graphs (and add letters denoting statistically important differences between means).
- The discussion is too long. Moreover the quality of discusssion is very low. Lines 185 – 213 contains elementary, school informations. The authors didn’t study the mechanism of herbicide action in C. sinensis therefore discussing the mechanisms of action is unjustified. The lines 235 – 245 about the risk for the operator are not justified by the research. The authors didn’t study the toxicity or risk for the operator.
- There is too high amount of positions in the references which are not scientific papers.

Author Response

(The authors gave the same response as above.)

Round 2
Reviewer 1 Report
The article has been improved during the reviewing process. Thus, this article can be accepted for publication on Plants.
Reviewer 3 Report
The manuscript has been improved by the authors therefore I recommend to publish it with some minor modifications.
lines 105 -118 and Figure 1 – this table together with description of weather data should be moved to Material and methods section. The weather observations haven’t been done by the authors. They have been done by Old Hidden Vale Station, Grandchester, therefore it can’t be placed in the result section.
Figure 2 – please add unit to the y axis
